# The Concept of Healthy Behaviours in Obesity May Have Unintended Consequences

**DOI:** 10.3390/nu15010012

**Published:** 2022-12-20

**Authors:** Hilary C. Craig, Zoë M. Doran, Carel W. le Roux

**Affiliations:** 1Diabetes Complications Research Centre, UCD Conway Institute of Biomedical and Biomolecular Research, School of Medicine, University College Dublin, Belfield, Dublin D04 VIW8, Ireland; 2Independent Researcher, Dublin D04 VIW8, Ireland

**Keywords:** obesity, pharmacotherapies, lifestyle therapies, self-care, integrated care, stigma, chronic disease management

## Abstract

Obesity has become a global epidemic, representing a major health crisis, with a significant impact both in human and financial terms. Obesity was originally seen as a condition, not a disease, which was considered self-inflicted. Thus, it was understandable that a simplistic approach, such as eat less and move more was proposed to manage obesity. Over the last 25 years, the perception of obesity has been gradually changing and the awareness has risen that it is a disease in its own right and not just a precipitating factor for type 2 diabetes, non-alcoholic fatty liver disease (NAFLD), etc. Creation of a comprehensive algorithm for the management of obesity needs to be informed by an in-depth understanding of the issues impacting the provision of treatment. Promotion of healthy behaviours is essential to help the population become healthier, but these are not obesity treatment strategies. Twenty percent of patients with obesity may respond to approaches based on healthy behaviour, but the 80% who do not respond should not be stigmatised but rather their treatment should be escalated. The unintended consequences of promoting healthy behaviours to patients with obesity can be mitigated by understanding that obesity is likely to be a subset of complex diseases, that require chronic disease management. Once the biology of the disease has been addressed, then healthy behaviours may play an invaluable role in optimising self-care within a chronic disease management strategy.

## 1. Obesity as a Set of Neurological Diseases and Not a Behaviour Problem

Obesity has become a global epidemic, representing a major health crisis, with a significant impact both in human and financial terms. Historically, obesity was seen to be a problem confined to high-income countries, but since 1975, the prevalence of obesity has tripled worldwide [1], accounting for at least, 2.8 million deaths annually, due to the associated complications of obesity. The morbidity and mortality attributable to obesity result in a huge burden on health care services [2] and the global cost annually, with OECD countries estimated to spend USD purchasing power parity (PPP) 311 billion every year to treat diseases caused by obesity [3]. The 2022 World Health Organisation (WHO) European report of Obesity found that 60% of adults have a BMI >25 kg/m^2^ [4]. This report also found that obesity rose by 21% between 2006 and 2016 and by 138% since 1975 in the WHO European region. Estimates suggest that obesity causes more than 1.2 million deaths across the European Region each year [4]. The incidence of the complications of obesity such as liver disease, diabetes and cancer is increasing, which substantially increases the negative impact on patients’ morbidity and mortality [2,3,5]. While recognising that the member states have made considerable progress in improving areas of prevention and lifestyle interventions to date, not a single member state of the WHO European Region has met their targets for stopping the rise in obesity and diabetes [4].

Obesity was originally seen as a condition, not a disease, which was considered self-inflicted. Thus, it was understandable that a simplistic approach, such as eat less and move more, was proposed to manage obesity. Over the last 25 years, the perception of obesity has been gradually changing and the awareness has risen that it is a disease in its own right and not just a precipitating factor for type 2 diabetes, NAFLD, etc. [6,7,8]. In 1997, the World Health Organisation recognised obesity as a chronic disease [9]. Subsequently, in March 2021 the European Union issued a brief in which it described obesity as a ‘chronic and relapsing disease’, which in turn acts as a gateway for other non-communicable diseases [6]. Obesity is now being considered as a possible set of neurological diseases, many of which develop over the course of a person’s life, with the environment, biology, genetic and behavioural factors all contributing [10]. Despite obesity being included in the International Classification of Diseases since 1948 [9], those suffering from obesity have long been under-served by the health care system and the management of these patients predominantly relies on changing lifestyles. Thus, the concept of encouraging healthy behaviours has been central in the pursuit of changing lifestyles. The incidence statistics indicate that this has not been a very effective strategy [4,11] and if we are to arrest this growing epidemic, the management paradigm must change.

Overnutrition and obesity are linked with abnormal functionality of homoeostasis brain networks. Neuroimaging data have been used to study underlying mechanisms and the consequences of obesity and diet at the level of the hypothalamus. BMI and visceral fat correlate with accelerated brain age, microstructure of the hypothalamus, lower thickness and connectivity in reward-related areas, and white matter lesion load. This is likely mediated through systemic low-grade inflammation. Some, but not all, longitudinal studies suggest beneficial effects of weight loss and healthy diets such as plant-based nutrients and dietary patterns on brain ageing and cognition [12,13]. Antioxidant activity correlates with prebiotic effects of oat phenolic compounds, which may yet be another avenue to explore during self-care [14]. The complex interplay of multiple metabolic, genetic, behavioural and environmental factors contributes to various subtypes of obesity, with environmental factors thought to be the proximate cause of the increased prevalence of obesity in recent decades [15,16]. For example, its rise in countries such as China is generally attributed to the change from an agriculturally centred society to and urban one. In response to the WHO and EU, obesity has been formally branded with the status of a non-communicable diseases [6,16]. The implication is clear: obesity should be treated like any other chronic disease and rather than focusing on one aspect such as lifestyle interventions, there is a need to take a holistic view and consider the potential subtypes of the disease [6]. Consequently, we see a greater understanding of the complexity of the management of people living with obesity and an increasing awareness of the need to ensure that there is access to treatment strategies, within a framework of multi-disciplinary care if and when needed.

## 2. Asking People with Obesity to Behave Better While Not Treating Their Disease Can Lead to Frustration and Stigma

There are a number of treatment modalities available to tackle this disease, ranging from lifestyle changes, nutritional therapies, exercise therapies, pharmacotherapies, and surgical therapies [17]. However, it is evident that the majority of patients find their options limited to the simplistic approach of lifestyle advice, generally overseen by their primary Health Care Provider, plus or minus evaluation by a dietician. The current management of obesity, globally, is inadequate to stem the tide of this health care crisis. There is an urgent need to re-evaluate management, determine the reasons for the limitations of current standards of care and implement management algorithms that will facilitate a more robust approach to treatment and access to all available interventions, as deemed necessary [4,18].

The major drivers for a simplistic treatment approach to obesity are economic, as reflected by health care policies, as well as the perception of obesity as a self-inflicted condition, not meriting the use of precious health care resources [5,19]. Historically, health care policies relating to obesity relied heavily on prevention, as Governments attempted to reduce obesity either through sugar taxes, education, or through improving the physical environments promoting healthy eating and activity [5]. The majority of health policies have relied on behavioural change to try to get people to change behaviours that can lead to obesity [19,20]. One could argue that this is primarily for economic reasons as behavioural interventions are less expensive than interventions requiring pharmacotherapy or surgery [21]. It should be noted that the implementation of policies that focus solely on individual responsibility can promote stigma by singling out people with the disease of obesity [21]. Obviously, in terms of the optimal strategies from a health policy perspective tackling people’s behaviour to eat more healthily and do more physical exercise can have a beneficial effect and has the benefit of being a low cost strategy for the exchequer [19]. However, as obesity subtypes often affect the sub cortical area of the brain, it can be argued that these strategies need to be implemented in combination with biological treatments to optimise the health gain for patients [18,22].

It is understandable that public health programmes take a utilitarian approach, the greatest good for the greatest number of people. They have to strike a balance between the collective wellbeing, an individual’s concerns and the costs of complications of diseases such as obesity [23]. Therein lies the attraction of public health policies that attempt to protect people from a disease through prevention, hence the primary focus to control obesity is on lifestyle interventions and healthy behaviours [23]. The emphasis is on the collective and social responsibility. However, this may translate to a paternalistic approach to prevent unhealthy behaviours, limiting people’s choices.

Non-communicable diseases such as obesity account for two-thirds of the disease burden globally and this is due to increase to three-quarters by 2030 [24]. Lifestyle interventions have merit, as with obesity education; 64% of patients show improvement in behaviours, 30% had no effect and 6% showed worse behaviour [25]. However, the ever-increasing incidence of obesity demonstrates that lifestyle interventions alone are not controlling the epidemic and their widespread acceptance as management could prevent more effective treatments and therapies being contemplated, as the lack of understanding about obesity can influence policies [18,23]. Puhl et al., 2010, discuss the notion that asking people with obesity to behave better while not treating their disease may not work [11]. They found that patients who have lost weight through lifestyle modification typically regain 30% to 35% of their lost weight during the year following treatment and regained most (if not all) of their weight within 5 years [11]. Thus, asking people to behave better, when the biological causes of obesity are not addressed, does not appear effective and may lead to the frustration of patients and clinicians.

The psychological impact on patients of persevering with behaviour advice without the disease of obesity being treated needs to be addressed. Clinician scientists are advocating for better treatment of obesity and the need to increase public awareness about the aetiology of obesity. However, it is clear that this narrative, while based in science, is not one that society or the media want to hear [11]. It is easier to blame the development of obesity as a personal failure or flaw. The disease can be very visual in nature and the judgement about its development not only stigmatises the person but can do considerable psychological damage. Stigmatising a person because of a disease is simply an ineffective strategy. As Puhl et al. state, ‘public health policy can either protect those afflicted with a disease from discrimination or can promote unfair treatment and disparities’ [11].

## 3. Once Obesity Is Effectively Treated, Patients Lose Weight without Effort

The key metric to evaluate the effectiveness of obesity management is health gain not weight loss. Healthy behaviours can result in weight loss but weight plateaus are inevitable and often weight is regained [26]. Consequently, there is a need to determine the utility of pharmacological and surgical management strategies. The semaglutide regulatory studies, STEP 1 and STEP 3, were conducted in adults with obesity, with one or more associated obesity complications, other than diabetes. The patients were randomised to receive semaglutide 2.4 mg or a placebo, administered once weekly, combined with a 500 kcal deficit diet and 150 min of exercise (in STEP 1) or intensive behavioural intervention (in STEP 3), over a period of 68 weeks [27,28]. The intensive behavioural therapy consisted of individual counselling sessions to support patients’ adherence to a low-calorie diet and to encourage them to increase their physical activity [27,28]. Both studies showed that after 68-week treatment period patients who received semaglutide achieved a sustainable, clinically relevant reduction in weight. The patients in STEP 1 with a less intensive behaviour program who were on placebo lost approximately 2% of their body weight, while those in STEP 3 with a very intensive behaviour program combined with placebo lost almost 8% of their weight. However, the patients in STEP 1 and STEP 3 lost similar amounts of weight (~16%) with semaglutide 2.4 mg despite very different intensity in behaviour programs [27,28]. When the same medications were used in children and adolescents in the STEP TEEN study, the placebo arm with a 500 kcal deficit diet and 150 min of exercise program lost no weight while the semaglutide 2.4 mg lost ~16% of their body mass index [29] similar to STEP 1 and STEP 3 [27,28]. These studies argue that the medication delivers the same amount of weight loss irrespective of the intensity of the behaviour programs that are paired with the medications. The management of dyslipidaemia is another example of once the biology of the disease has been addressed then healthy behaviours may play an invaluable role in optimising self-care within a chronic disease management strategy. For example, there now appears synergism of how whole grain can benefit the management of dyslipidaemia via gut microbiota [30].

Treatment options such as nutritional therapies, pharmacotherapies, or surgical therapies which result in more than 10% weight loss improve health in those with obesity [31]. Although several factors such as the environment, economic and social disparities contribute to obesity, it is harder to show that removing these factors are good treatments for the disease. Thus, although the disease may be caused by factor A, the removal of factor A may not reverse the disease, but rather treatment B may be needed. Whatever works, it does appear that obesity requires chronic disease management.

Chronic disease management requires an integrated approach to prevent and manage chronic illness with the emphasis invariably on self-management. The most frequent chronic care model used is the self-management support model which has demonstrated improvement in conditions such as hypertension and diabetes [32]. Norris S et al. identified the component interventions, involved in chronic disease management. Chronic disease management is population based, specifically targeted to each disease, should include implementation of appropriate clinical guidelines, determination and implementation of targeted health care interventions, use of clinical information systems and measurement and management of outcomes [33]. Patients may be managing more than one chronic illness and will require a coordinated management approach on all aspects of each disease, as such a proactive multi-component service is required from a multi-disciplinary team [33]. Ongoing interaction with health care professionals has shown to improve the management of obesity with longer positive outcomes [26].

## 4. Self-Care, a Critical Strategy to Enable Chronic Disease Management

Self-care as part of chronic disease management has been defined as people choosing a healthy lifestyle, managing their own social, emotional and psychological wellbeing, and managing long-term conditions to prevent further illness [34]. This should not be confused with the self-management of lifestyle interventions. Self-care, thus defined, could prove to be a valuable additional step in achieving control of chronic disease. Successful programmes need a whole system approach as it is not achieved by patients acting in isolation [34]. Despite multiple resources allocated to promoting self-care, current research show only modest effects in respect to social activities, managing stress, preparing healthy foods and participating in exercise [35]. Better support for self-care where health professionals take a more personal approach has better results [35].

The challenges with self-care include the level of understanding of the patient, access to care, good communication with health care professionals as well as support about changing beliefs to improve people’s ability to self-manage [36]. Reigel et al., 2021, reviewed 9309 citations with 233 studies included in their final review, which also included randomised controlled trials comparing behavioural or education self-care intervention. Nine chronic conditions were included [37]. They discovered that there were major gaps in self-care interventions including a lack of psychological support as well as behaviour change techniques [37]. However, general practitioners that support patients with self-care for chronic disease management have improved outcomes, indicating that this additional management step is one worthy of further investigation [38].

## 5. Integrating Care

Addressing weight stigma, discrimination, genetics, education and treatment all have a role to play in dealing with an epidemic of obesity that is largely using a blame narrative to manage it, to no avail [7,21]. We need a societal change to look at the science of the disease and provide not just prevention programmes but chronic treatment programmes to contain this epidemic [7]. The first line management option remains lifestyle interventions alone and can be effective in 20% of cases but for those patients who do not respond to the treatment the management algorithm should include escalation to medication and/or surgery if indicated [39]. Obesity is a multifactorial disease that requires a multifactorial response [7]. While treatment options and health behaviours can help manage the disease it is essential for the individual to participate in self-care, it is time for patients’ voices to be heard. Self-care in relation to obesity often refers to the prevention of relapse of obesity encouraging the population to choose healthy behaviours. Self-care refers to information, education on healthy options, being more active and lifestyle changes and it has a very important role in the prevention of relapse of obesity and indeed in ensuring the disease of obesity remains optimally controlled [40]. Nonetheless patients’ needs must be assessed, and the appropriate health care providers identified.

Patients want more information, education and support from their health care provider to help them manage their chronic disease. Patients want to be engaged on a regular basis to support them in managing their condition [32]. Self-care support in chronic disease management results in significant improvements in disease outcomes [32]. Participating in healthy behaviours to help manage the disease is essential not to create more weight loss or even weight loss maintenance, but to ensure the disease of obesity remains optimally controlled.

Creation of a comprehensive algorithm for the management of obesity needs to be informed by an in-depth understanding of the issues impacting the provision of treatment. The level of understanding of health care professionals in respect to the management of obesity needs to be established.The common practice of confining management to lifestyle interventions needs to be challenged.Are there barriers to accessing interventional treatments such as medication or surgery?Is cost an issue?How can we ensure a multi-disciplinary approach to care if and when needed to include, dietary, lifestyle, clinical, surgical and psychological support?How do we involve patients more to understand their perceptions, wishes, fears and experiences to date?

Patients often feel that they are not consulted or listened to concerning their care, that their obesity is still considered a self-inflicted condition to be managed conservatively by diet and exercise. Access to other alternatives is limited.

## 6. Conclusions

In conclusion, promotion of healthy behaviours is essential to help the population become healthier, but these are not obesity treatment strategies. Twenty percent of patients with obesity may respond to approaches based on healthy behaviour, but the 80% who do not respond should not be stigmatised but rather their treatment should be escalated [39]. The unintended consequences of promoting healthy behaviours to patients with obesity can be mitigated by understanding that obesity is likely to be a subset of complex diseases that require chronic disease management. Once the biology of the disease has been addressed, then healthy behaviours may play an invaluable role in optimising self-care, within a chronic disease management strategy.

## Data Availability

Not applicable.

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
