# Peer review of "The Concept of Healthy Behaviours in Obesity May Have Unintended Consequences"

_nutrients, 2022, doi:10.3390/nu15010012_

Round 1

Reviewer 1 Report

Obesity has become a global epidemic, representing a major health crisis, with a significant impact both in human and financial terms. Obesity was originally seen as a condition, not a disease, which was considered self-inflicted. Thus, it was understandable that a simplistic approach, such as eat less and move more was proposed to manage obesity.

1.      The unintended consequences of promoting healthy behaviors to patients with obesity can be mitigated by understanding that obesity is likely to be a subset of complex diseases, that require chronic disease management. Once the biology of the disease has been addressed then healthy behaviors may play an invaluable role in optimizing self-care within a chronic disease management strategy. In fact, the conception should be hyperlipidemia or metabolic syndrome, please reference and cite relevant references (Whole grain benefit: oat β-glucan and phenolic compounds synergistically regulates hyperlipidemia via gut microbiota in high-fat-diet mice. Food & Function, 2022, Doi: 10.1039/d2fo01746f.).

2.      In fact, obesity has inflammation, which can be neurological diseases. Diet factor play an important role in regulating the obesity.  Please search and and cite relevant reference.

3.      The behavior is effect way, in meanwhile the wholegrain consumption and gut microbiota also play an important role in regulating the obesity (The positive correlation of antioxidant activity and prebiotic effect about oat phenolic compounds. Food Chemistry, 402(2023): 134231.)

4.      The data about obesity should be analyzed in table or figure.

Author Response

Comment 1. Obesity has become a global epidemic, representing a major health crisis, with a significant impact both in human and financial terms. Obesity was originally seen as a condition, not a disease, which was considered self-inflicted. Thus, it was understandable that a simplistic approach, such as eat less and move more was proposed to manage obesity.

The unintended consequences of promoting healthy behaviors to patients with obesity can be mitigated by understanding that obesity is likely to be a subset of complex diseases, that require chronic disease management. Once the biology of the disease has been addressed then healthy behaviors may play an invaluable role in optimizing self-care within a chronic disease management strategy. In fact, the conception should be hyperlipidemia or metabolic syndrome, please reference and cite relevant references (Whole grain benefit: oat β-glucan and phenolic compounds synergistically regulates hyperlipidemia via gut microbiota in high-fat-diet mice. Food & Function, 2022, Doi: 10.1039/d2fo01746f.).

Response 1. We agree with the reviewer that the previous approach of “eat less and move more“ appeared to be a logical step to the problem given our understanding at the time. Now that the scientific understanding of the pathophysiology of obesity has changed it also allows us to adapt our approach. We have added on page 4 line 174: “The management of dyslipidaemia is a good example of once the biology of the disease has been addressed then healthy behaviours may play an invaluable role in optimizing self-care within a chronic disease management strategy. For example there now appears synergism of how whole grain can benefit the management of dyslipidaemia via gut microbiota (REF. Food & Function, 2022, Doi: 10.1039/d2fo01746f. https://pubs.rsc.org/en/content/articlelanding/2022/fo/d2fo01746f)

Comment 2.      In fact, obesity has inflammation, which can be neurological diseases. Diet factor play an important role in regulating the obesity.  Please search and and cite relevant reference.

Response 2. We agree with the reviewer and we have added a sentence on page 2 line 64-71: “Overnutrition and obesity are linked with abnormal functionality of homoeostasis brain networks. Neuroimaging data have been used to study underlying mechanisms and the consequences of obesity and diet at the level of the hypothalamus. BMI and visceral fat correlate with accelerated brain age, microstructure of the hypothalamus, lower thickness and connectivity in reward-related areas, and white matter lesion load. This is likely mediated through systemic low-grade inflammation. Some, but not all longitudinal studies suggest beneficial effects of weight loss and healthy diets such as plant-based nutrients and dietary patterns on brain ageing and cognition (REF: https://pubmed.ncbi.nlm.nih.gov/36345149/ , https://pubmed.ncbi.nlm.nih.gov/36251886/ , https://pubmed.ncbi.nlm.nih.gov/35681242/ , https://pubmed.ncbi.nlm.nih.gov/34621773/ )

Comment 3.      The behavior is effect way, in meanwhile the wholegrain consumption and gut microbiota also play an important role in regulating the obesity (The positive correlation of antioxidant activity and prebiotic effect about oat phenolic compounds. Food Chemistry, 402(2023): 134231.)

See Carel

Response 3. The reviewer raises an important point and we have added on page 2 line 71: “Antioxidant activity correlates with prebiotic effects of oat phenolic compounds, which may yet be another avenue to explore during self-care (REF: https://www.sciencedirect.com/science/article/abs/pii/S0308814622021938 )

Comment 4.      The data about obesity should be analyzed in table or figure.

Response 4. We would be grateful for more clarification, because we are not clear on what data we need to tabulate. We would be grateful for the guidance of the Editor because we think data on obesity is outside of the remit of our manuscript.

Reviewer 2 Report

    This short review highlights the importance and need for a chronic management strategy, including treatment for obesity, except for promoting healthy behaviors. It is an interesting review. Some comments below are for the authors to consider for revision.

1.     In the review title, ‘How…’ is emphasized. But it seems the article does not provide much relevant information. It would be better if the title could be more straightforward.

2.     The article was not for the review of pharmacotherapies for obesity. At least ‘surgery’ may not be a keyword.

3.     Suggest that the full name of ‘NAFLD,’ line 15, and ‘PPP,’ line 35, page one, should be given.

4.     The abbreviation of WHO may not be defined twice, line 52, page 2; and it is not necessary for HCP, line 82, page 2, and RCTs, line 196, page 4.

5.     Suggest a reference should be provided for the statement ‘ obesity is now being … all contributing’, ling55-57, page 2.

6.     In the Abstract, Integrating care, and Conclusion, the authors indicate that 20% of obesity patients may respond, but 80% may not respond to the healthy behavior change approach. This is not the review conclusion; at least the reference(s) in relevant sections should be provided.

7.     Please check the writing or grammar, e.g., lines 59, 68, 91, 102, and 113.

8.     Author contributions: Please remove wordings on the journal’s instructions.

9.     Please go through the reference list, and revise references when needed.

Author Response

This short review highlights the importance and need for a chronic management strategy, including treatment for obesity, except for promoting healthy behaviors. It is an interesting review. Some comments below are for the authors to consider for revision.

Comment 1. In the review title, ‘How…’ is emphasized. But it seems the article does not provide much relevant information. It would be better if the title could be more straightforward.

Response 1. We agree with the reviewer and we have changed the title to: The concept of healthy behaviours in obesity may have unintended consequences.

Comment 2.  The article was not for the review of pharmacotherapies for obesity. At least ‘surgery’ may not be a keyword.

Response 2. As requested, we have deleted the word surgery from the keywords.

Comment 3.  Suggest that the full name of ‘NAFLD,’ line 15, and ‘PPP,’ line 35, page one, should be given.

Response 3. We apologise for the oversight. We have now defined all terms including Non-alcoholic fatty liver disease and purchasing power parity (PPP) prior to using abbreviations.

Comment 4.     The abbreviation of WHO may not be defined twice, line 52, page 2; and it is not necessary for HCP, line 82, page 2, and RCTs, line 196, page 4.

Response 4. We apologise for the oversight. We have now defined all terms only once prior to using them.

Comment 5.     Suggest a reference should be provided for the statement ‘obesity is now being … all contributing’, ling55-57, page 2.

Response 5. As requested, we have provided a reference for the statement. Gómez-Apo E, Mondragón-Maya A, Ferrari-Díaz M, Silva-Pereyra J. Structural Brain Changes Associated with Overweight and Obesity. Journal of Obesity. 2021;2021:6613385.

      Comment 6.     In the Abstract, Integrating care, and Conclusion, the authors indicate that 20% of obesity patients may respond, but 80% may not respond to the healthy behavior change approach. This is not the review conclusion; at least the reference(s) in relevant sections should be provided.

      Response 6. As requested we have provided the references in the manuscript on page 5 line 230, page 6 line 268, indicating that approximately 20% of patients achieve more than 10% weight loss with a healthy behaviour change approach (REF https://pubmed.ncbi.nlm.nih.gov/23796131/ and https://pubmed.ncbi.nlm.nih.gov/30852132/ )

      Comment 7.     Please check the writing or grammar, e.g., lines 59, 68, 91, 102, and 113.

      Response 7. We apologise for the mistakes. We have now carefully checked lines 59, 68, 91, 102, and 113 and have corrected all the grammatical errors we could find.

Comment 8.  Author contributions: Please remove wordings on the journal’s instructions.

Response 8. We have made the changes as requested

Comment 9.  Please go through the reference list, and revise references when needed.

Response 9. We have made the changes as requested

Round 2

Reviewer 1 Report

The author has revised the manuscript as the suggestion of the reviewer. It will be better to check the minor revision.

Reviewer 2 Report

Thank you for considering my comments and revisions.

Reference formats need to be checked carefully in the final proofreading.